

# Analysis of ionospheric structure influences on residual ionospheric errors in GNSS radio occultation bending angles based on ray tracing simulations

**Congliang Liu[1,3], Gottfried Kirchengast[2,3,1,5], Yueqiang Sun[1,3,4], Kefei Zhang[5], Robert Norman[5], Marc Schwaerz[2,3], Weihua Bai[1,3,4], Qifei Du[1,3], and Ying Li[6]**

[1] National Space Science Center, Chinese Academy of Sciences (NSSC/CAS) and Beijing Key Laboratory of Space Environment Exploration, Beijing, China

[2] Wegener Center for Climate and Global Change (WEGC) and Institute for Geophysics, Astrophysics, and Meteorology/Institute of Physics, University of Graz, Graz, Austria

[3] Joint Laboratory on Occultations for Atmosphere and Climate (JLOAC) of NSSC/CAS, Beijing, China, and University of Graz, Graz, Austria

[4] University of Chinese Academy of Sciences, Beijing, China

[5] SPACE Research Centre, RMIT University, Melbourne, VIC, Australia

[6] Institute of Geodesy and Geophysics (IGG), Chinese Academy of Sciences, Wuhan, China

Correspondence to: Congliang Liu (E-mail: liucongliang1985@gmail.com)

## Abstract

The Global Navigation Satellite System (GNSS) radio occultation (RO) technique is widely used to observe the atmosphere for applications such as numerical weather prediction and global climate monitoring. The ionosphere is a major error source to RO at upper stratospheric altitudes and a linear dual-frequency bending angle correction is commonly used to remove the first-order ionospheric effect.



However, the residual higher-order ionospheric error (RIE) can still be significant so that it needs to be further mitigated for high accuracy applications, especially from 30 km altitude upward where the RIE is most relevant compared to the decreasing magnitude of the atmospheric bending angle. In a previous study we quantified RIEs using an ensemble of about 700 quasi-realistic end-to-end simulated RO events, finding typical RIEs at the 0.1 to 0.5 μrad noise level, but were left with 26 exceptional events with anomalous RIEs at the 1 to 10 μrad level that remained unexplained. In this study, we focused on investigating the causes of the high RIE of these exceptional events, employing detailed along-raypath analyses of atmospheric and ionospheric refractivities, impact parameter changes, and bending angles and RIEs under asymmetric and symmetric ionospheric structures. We found that the main causes of the high RIEs are a combination of physics-based effects, where asymmetric ionospheric conditions play the primary role, more than the ionization level driven by solar activity, and technical ray tracer effects due to occasions of imperfect smoothness in ionospheric refractivity model derivatives. We also found that along-ray impact parameter variations of more than 10 to 20 m are well possible due to ionospheric asymmetries, and depending on prevailing horizontal refractivity gradients are positive or negative relative to the initial impact parameter at the GNSS transmitter. Furthermore, mesospheric RIEs are found generally higher than upper stratospheric ones, likely due to being closer in tangent point heights to the ionospheric E layer peaking near 105 km, which increases RIE vulnerability. In future we will further improve the along-ray modeling system to fully isolate technical from physics-based effects and to use it beyond this work for additional GNSS RO signal propagation studies.

## 1 Introduction

Global Navigation Satellite System (GNSS) radio occultation (RO) (Melbourne et al., 1994; Kursinski et al., 1997; Hajj et al., 2002) is a relatively new atmospheric sounding technique. It can deliver data traceable to the international standard of time (the SI second) and has a demonstrated capacity for



monitoring decadal-scale climate change in the free atmosphere (Steiner et al., 2009, 2011, 2013; Anthes, 2011; Foelsche et al., 2011; Lackner et al., 2011; Ho et al., 2012; Angerer et al., 2017). This capacity rests on RO's unique combination of characteristics such as high vertical resolution, high accuracy, long-term stability and global coverage (Kursinski et al., 1997; Anthes, 2011; Steiner et al., 2011). Figure 1 illustrates the GNSS radio occultation geometry that constitutes the basis of the RO technique. The focus is to schematically show essential aspects relevant to this study on along-ray ionospheric influences on RO bending angles, which deepens insight on top of our recent Liu et al. (2015) study.

Ionospheric error is significant in GNSS RO observations (e.g., Kursinski et al., 1997; Mannucci et al., 2011; Liu et al., 2013) and a dual-frequency linear combination of RO bending angles is usually implemented to correct for the first-order ionospheric effect (Vorob'ev and Krasil'nikova, 1994; Ladreiter and Kirchengast, 1996). However, the higher-order residual ionospheric error (RIE) after this correction is still not negligible for high accuracy applications such as RO-based climate change monitoring (Steiner et al., 2011, 2013). This applies especially above about 35 km altitude where the RIE becomes increasingly relevant compared to the exponentially-decreasing magnitude of the neutral atmospheric bending angle (Syndergaard, 2000; Mannucci et al., 2011; Danzer et al., 2013, 2015; Liu et al., 2013, 2015; Healy and Culverwell, 2015).

Moreover, the RIE can propagate downwards into the lower stratospheric retrievals of refractivity and temperature through the Abel integral and the hydrostatic integral (Kursinski et al., 1997; Gobiet and Kirchengast, 2004; Steiner and Kirchengast, 2005; Gobiet et al., 2007). It is therefore essential to better understand and further mitigate the RIE in order to enable benchmark-quality stratospheric RO retrievals.

A wide array of studies related to a better understanding of higher-order ionospheric errors in GNSS RO data have been conducted already by a range of scientists (Bassiri and Hajj, 1993; Vorob'ev and Krasil'nikova, 1994; Ladreiter and Kirchengast, 1996; Syndergaard, 2000; Gorbunov, 2002; Hoque and Jakowski, 2010, 2011; Mannucci et al., 2011; Danzer et al., 2013, 2015; Healy and Culverwell, 2015).



A few of these also suggested ways of correcting higher-order RIEs in RO bending angles (Syndergaard, 2000; Danzer et al., 2013; Healy and Culverwell, 2015), which may be applied on top of the standard dual-frequency correction introduced by Vorob'ev and Krasil'nikova (1994).

The convenient formulation introduced by Healy and Culverwell (2015), which adds a fairly simple higher order squared-bending angle difference term to the standard correction, is meanwhile applied in operational processing of the data from the European MetOp (Meteorological Operational Satellites) RO mission (Luntama et al., 2008; C. Marquardt, EUMETSAT Darmstadt, pers. communications, 2017). Recently, Angling et al. (2017) further improved the empirical modeling of the "kappa coefficient" in this formulation, by accounting for solar zenith angle, solar flux (F10.7 index) and altitude dependencies.

In our work over the recent years we have assessed the variation of bending angle RIEs (biases and standard deviations) with solar activity, latitudinal region, and with or without the assumption of ionospheric spherical symmetry and of co-existing RO observing system errors, using quasi-realistic end-to-end simulations for single RO events (Liu et al., 2013) and a full-day ensemble of RO events (Liu et al., 2015). As shown in these explanatory simulation studies, in overall agreement with the empirical study of Danzer et al. (2013), the RIE biases have a clear negative tendency and a bias magnitude increasing with solar activity. as well as are somewhat affected by ionospheric spherical symmetry or asymmetries. They also are markedly increased both by increasing solar activity and ionospheric asymmetries.

What remained unexplored in our Liu et al. (2015) study and was also not yet explored elsewhere—but is critical to be understood for further improvement of the existing RIE corrections that apply spherical symmetry (Syndergaard, 2000; Healy and Culvervell, 2015; Angling et al., 2017)—are the influences of the three-dimensional and asymmetric ionospheric structures along the GNSS-to-LEO signal paths on the RIE, in particular the conditions that may lead to anomalously high RIEs.





A first step in this direction, though not focusing on bending angle RIEs, was the study by Mannucci et al. (2011) which found that under ionospheric storm conditions anomalous effects can be significant. Another step was the somewhat puzzling side result in our Liu et al. (2015) study that the end-to-end simulations of an ensemble of about 700 RO events produced about two dozen RIE outlier profiles. The basis were 3D ray tracing simulations, where the ionospheric model NeUoG (Leitinger and Kirchengast, 1997) was used as quasi-realistic model for large-scale 3D ionospheric structures, together with the atmospheric model MSIS-90 (Hedin, 1991) for simple but representative neutral atmosphere reference conditions. More precisely, the RIE standard deviation of 26 profiles from the simulations exceeded a threshold value of 7 μrad within the upper stratosphere and mesosphere. These were therefore rejected from the ensemble statistics results reported by Liu et al. (2015).

In this study we now take focus on these 26 exceptional profiles and, by way of detailed along-ray analyses of ray tracing simulations, aim to shed light on the causes of anomalously high RIEs, with the goal to also deepen quantitative insight into how RIEs accumulate during signal propagation, along with accumulation of the total (atmospheric) bending angles that are the desired RO observables. In Sect. 2, the exceptional RO events and the simulation setup for exploring their bending angle RIEs are introduced. Section 3 provides the results, which we mainly discuss through detailed inspection of example events. Summary and conclusions are finally given in Sect. 4.

## 2 Exceptional RO events and investigation methodology

### 2.1 Exceptional RO events

The ensemble of RO events used by Liu et al. (2015) was simulated for 15 July 2008, adopting the European MetOp RO mission as example low Earth orbiter (Edwards and Pawlak, 2000), specifically thinking of MetOp-A that was launched as the first of the MetOp series in late 2006 (Luntama et al.,



2008). Each MetOp satellite is a sun-synchronous LEO satellite at about 820 km with the Global Positioning System (GPS) Receiver for Atmospheric Sounding (GRAS) on board (Loiselet et al., 2000) that acquires about 700 RO events per day (Luntama et al., 2008).

Using, as summarized above, simple spherically symmetric neutral atmospheric modeling (by MSIS-90) combined with 3D ionospheric modeling (by NeUoG), we simulated in that study the ensemble of daily RO events for 14 different end-to-end simulation cases. These included without-ionosphere (wi) cases as well as spherical symmetry (ss) and non-spherical symmetry (ns) ionospheric cases for low, medium, and high solar activity levels, under the assumption of either perfect observing system (op) with no errors or realistic observing system (or) with MetOp-type errors; for details see Liu et al. (2015), Table 2 and Sect. 2.3 therein. The total number of the simulated RO events found for the day was 723, of which 26 exceptionally noisy ones were classified as outliers (estimated bending angle RIE exceeding 7 μrad somewhere within 30 to 80 km). These 26 events are investigated closer in this study.

Figure 2a shows the global distribution of mean tangent point locations of all 723 events (as small triangles) and highlights the locations of the 26 exceptional events (as red triangles). The latter appear to cluster in their majority (18 of the 26) over the European Asian and Indian Ocean regions (EAC and IOC, highlighted as boxes); the remaining 8 events are cluttered more diversely in other extratropical locations, mainly in the northern hemisphere. Figures 2b and 2c depict the RIE bias and standard deviation estimated for the upper stratosphere and mesosphere (30-80 km) for the 26 events, for the non-spherical symmetry ("opns") and spherical symmetry ("opss") ionospheric conditions, respectively. Intercomparing Fig. 2b and 2c clearly shows that the main driver of anomalously high RIEs are asymmetric ionospheric conditions as only few events (6 of the 26) exhibit large RIE standard deviations (exceeding 1 μrad) also in case of symmetric ionospheric conditions.

Related to the clusters, one can see that, in the "opss" case, almost all noisy exceptional events occurred in the IOC, while in the "opns" case the noisiest ones occur in both the EAC and IOC. Related to solar



activity, one can see that in both the "opss" and "opns" cases higher ionization (F10.7) levels generally lead to increased RIEs, compared to lowest ionization (F10.7 = 70), but the picture is ambiguous and often also medium solar activity leads to higher RIEs than high solar activity.

These overall characteristics revealed by Fig. 2 point, in particular, to two facts that shall guide our detailed investigation for better understanding of anomalous RIEs, 1.) asymmetric ionospheric conditions play a key role, more than ionization levels and possible geographic location dependencies, and so inspection of the along-ray signal dynamics is essential, 2.) the several exceptions from the overall characteristics, and some geographic clustering that has no obvious physics-based explanation, indicate that there is no single clear cause for the anomalous RIEs and that some perturbations may also come in from the technical challenge of smooth ray tracing at millimetric excess phase accuracy through 3D ionospheric models like NeUoG.

We inspected the bending angle RIE profiles of the 26 events over the 20 to 80 km height range, including also their underlying excess phase RIE profiles, and chose three representative events that we will explore in detail below for improved RIE insight: an extremely noisy event (Occ.530 from the EAC) and a medium noisy event (Occ.20 from the IOC) from the 26 exceptional events, both used at medium solar activity, and a reference event from the 697 standard events, with low-noise RIE (Occ.25). Table 1 summarizes the main parameters for these three events and Figure 3 illustrates them in terms of excess phases, bending angles, and the associated RIEs.

Figure 3a shows the behavior of the excess phases of the three events. The L1 and L2 excess phases are around −11 to −15 m and −18 to −25 m, respectively, typical for medium solar activity (Liu et al., 2013). After the standard ionospheric correction the Lc excess phases are found near 0 m as should be the case. The excess phase RIE profiles (Fig. 3b) exhibit some spiky behavior for the two exceptional events, on top of comparatively low-noise RIEs otherwise. This points to unphysical values at the spiky impact height levels, given that the large-scale 3D ionospheric structure of the NeUoG model should be



physically unable to induce such sharp changes. It hence indicates that the ray tracing is technically challenged along the signal propagation paths pertaining to these levels by slight ionospheric model discontinuities which render millimetric excess phase accuracy unattainable for these ray paths.

As Leitinger and Kirchengast (1997) describe, substantial empirical modeling effort went into strict smoothness of the NeUoG electron density field and its 3D derivatives that are key for high-accuracy ray tracing; nevertheless some slight discontinuities have likely remained in some rare locations of the modeling space spanned by altitude, latitude, longitude, (universal) time, month, and solar activity. It will therefore be important to separate such technical modeling effects from the physical effects on the propagating signals that cause high RIEs.

Figure 3c shows that, for all three events, the difference between $\alpha_1$ and $\alpha_2$ somewhat increases with increasing impact height, a feature already visible in the Liu et al. (2013) results. It roots in the increasing ionospheric influence when tangent point heights gradually approach ionospheric E layer heights around 105 km from below. These overall differences between $\alpha_1$ and $\alpha_2$ amount to about 15 to 20 μrad near 80 km and are effectively eliminated by the standard ionospheric correction, bringing the $\alpha_c$ profile to near zero as should be the case. Nevertheless, substantial waveform-like perturbations remain on $\alpha_c$ for the two exceptional events Occ.530 and Occ.20, which even more clearly show up in the bending angle RIE profile (Fig. 3d).

Intercomparing Fig. 3d with Fig. 3b suggests that these waveform-like perturbations in the bending angle RIE are mainly induced by propagating the spiky excess phase perturbations through the bending angle retrieval, which involves filtering and a derivative operation from excess phase to Doppler shift (Schwarz et al., 2017). One main cause that has driven many of the exceptional events into the outlier range (i.e., into exceeding 7 μrad somewhere within 30 to 80 km) is thus evidently the technical effects from the ray tracing through the NeUoG ionosphere that is not perfectly smooth everywhere in its electron density and hence refractivity field derivatives. It is thus important to more closely explore the





along-ray signal dynamics in order to understand how such technical effects may occur along ray paths but in particular in order to better understand the physical effects that drive high RIEs. Our related along-ray analysis methodology is introduced next.

## 2.2 Investigation methodology

### 2.2.1 Ray tracing method

The ray tracing technique is commonly used for calculating propagation paths of an electromagnetic signal in a medium specified by a position-dependent refractive index field. It has become a significant tool for investigating signal propagation in RO technology. For example, Ladreiter and Kirchengast (1996), Syndergaard (2000), Gobiet and Kirchengast (2004), Steiner and Kirchengast (2005), Hoque and Jakowski (2010), Mannucci et al. (2011), Danzer et al. (2013, 2015), and Liu et al. (2013, 2015) have employed this method *inter alia* or with a main topical focus to investigate the ionospheric effects on GNSS RO signals.

For this study, the 3D numerical ray tracing technique integrated in the End-to-end GNSS Occultation Performance Simulation and Processing System version 5.6 (EGOPS 5.6) (Fritzer et al., 2013) was employed in the same way as by Liu et al. (2013; 2015) for simulating the GPS-to-LEO signal propagation through the atmosphere-ionosphere system; for a detailed description of the end-to-end simulation setup the reader is therefore referred to these recent studies. Here we specifically refined and enhanced this setup in the 3D ray tracing part by adding the co-computation and result extraction for a range of key variables along the propagation paths, instead of only providing the final observational variables of an RO event at the LEO receiver orbit.



## 2.2.2 Investigated variables

We implemented detailed along-ray diagnostic capabilities into the 3D ray tracer of the EGOPS 5.6 software (Fritzer et al., 2013), which is an extensively proven high-accuracy ray tracer originally developed in the 1990s (Syndergaard, 1998; 1999). In particular, we computed the following key diagnostic variables for all individual numerical steps along the ray paths simulated for the GPS L1 and L2 frequencies as well as for a reference case without ionosphere (Lr), with each ray path starting at the GPS transmitter position and ending at the LEO receiver position:

3D position in the WGS84-based Earth-centered Earth-fixed (ECEF) system, storing both the cartesian (X, Y, Z) and geodetic (height, latitude, longitude) coordinates; along-ray distance relative to the tangent point (TP), the latter evaluated as the ray's point of closest approach to the WGS84 ellipsoidal surface (parabolic vertex fit to the three along-ray positions closest to surface); atmospheric refractivity; L1 and L2 ionospheric refractivity; L1 and L2 impact parameter and impact parameter difference to the initial impact parameter at the GPS transmitter position (termed "delta impact parameter", induced along the ray in case of non-spherical symmetry conditions); accumulated L1, L2, and ionosphere-corrected (Lc) bending angle (bending angle accrued from the GPS transmitter position to the along-ray position); and residual ionospheric error (RIE) of the Lc bending angle, estimated relative to the Lr bending angle obtained from a simulation case without ionosphere (Liu et al., 2013).

These along-ray variables are computed for all available ray paths from 80 km to 20 km impact height, which are produced at 50 Hz sampling rate for any RO event investigated. This leads to a dense sampling by roughly around 1500 ray paths in this altitude range (i.e., typical average scan velocities of RO events are near 2 km s$^{-1}$ in this domain). Likewise the ray tracer provides fairly dense along-ray stepping, employing an adaptive step size concept with finest steps at highest local refractive index changes (for details see, e.g., Syndergaard, 1999; Fritzer et al., 2013). Together these features enable to inspect the propagation characteristics of RO events through the atmosphere-ionosphere system at very





high resolution in a convenient 2D along-ray distance vs. impact height coordinate system that accurately represents the real 3D-warped occultation event plane between the GPS and LEO orbit arcs.

We will inspect the results for the three representative RO events chosen (Occ.530, Occ.20, Occ.25; see Sect. 2.1 above) in this along-ray distance vs. impact height coordinate system. Before turning to this,
we briefly summarize here the equations for the along-ray computation of those key variables that we will inspect closely. This aims to facilitate an appropriate understanding and interpretation of the results.

On the basis of Snell's law, when the Earth's atmosphere and ionosphere is assumed spherically symmetric, Bouguer's rule can be used to describe the refraction of a ray path in terms of a constant
impact parameter (e.g., Budden, 1985),

$$a = nr\sin\Phi = \text{constant},\qquad(1)$$

where $a$ is the impact parameter, $r$ is the radial distance from the center of curvature of the refracted ray to any point of the ray path, $n$ is the refractive index (at radial distance $r$) which is related to refractivity $N$ via $n = 1 + 10^{-6} N$, and $\Phi$ is the local angle between the radial position vector and the ray direction
at any point of the ray.

Eq. (1) implies that, at each point along the ray path, the impact parameter $a$ is equal to its initial value at the GPS transmitter position in case of spherical symmetry, which leads to

$$a_i = n_i r_i \sin\Phi_i = (1 + 10^{-6} N_i)\left|\vec{R}_i \times \hat{r}_i\right| = \left|\vec{R}_G\right|\sin\Phi_G,\qquad(2)$$

where index $i$ counts the (numerical ray tracer) points along the ray path starting at the GPS transmitter
position $\vec{R}_G$ and ending at the LEO, $\vec{R}_i$ and $\hat{r}_i$ are the radial position vector and unit vector along the ray direction at point $i$, respectively, and $\Phi_G$ is the local angle between position vector and (initial) ray direction at the GPS transmitter where we can furthermore assume that the refractivity is zero.





In reality non-spherical symmetry conditions of appreciable size will frequently occur, in particular between the ionospheric signal propagation inbound from the GPS and (after propagating through the atmosphere at tangent heights below 80 km) the one outbound to the LEO (cf. Fig. 1); see, e.g., the RO events discussed by Liu et al. (2013). In order to inspect the impact parameter changes along the ray path in these cases where $a_i$ computed according to the 2$^{nd}$ R.H.S. term of Eq. (2) will vary along the ray path, we co-compute the delta impact parameter $\Delta a_i$ as the difference of the impact parameter at points $i$ of the ray path and the impact parameter $a_G$ at the GPS location,

$$\Delta a_i = a_i - a_G = \left(1 + 10^{-6} N_i\right)\left|\vec{R}_i \times \hat{r}_i\right| - \left|\vec{R}_G\right|\sin\Phi_G . \qquad (3)$$

Complementary to the geometrical parameters available from the ray tracing, the refractivity $N$ comprises atmospheric and ionospheric terms which are formulated based on standard equations as (e.g., Liu et al., 2015; Eqs. 1 and 4 therein),

$$N = N_{atm} + N_{ion} = C_{atm} \cdot p/T - C_{ion} \cdot N_e / f^2 \qquad (4)$$

where $C_{atm} = 77.60$ K hPa$^{-1}$ and $C_{ion} = 40.31 \cdot 10^6$ m$^3$ s$^{-2}$ are the classical refractivity coefficients, $p$ [hPa] and $T$ [K] are atmospheric pressure and temperature (modeled by MSIS-90), $N_e$ [m$^{-3}$] is the ionospheric electron density (modeled by NeUoG), and $f$ [Hz] is the GPS signal frequency ($f_{L1} = 1.57542$ GHz; $f_{L2} = 1.22760$ GHz).

In addition, the accumulated bending angle $\alpha_i$, that accrues from the GPS position to any point $i$ along the ray path, can be readily computed as the angle between the initial ray direction (unit vector $\hat{r}_G$) and the ray direction at point $i$ (unit vector $\hat{r}_i$),

$$\alpha_i = \arccos\left(\hat{r}_G \cdot \hat{r}_i\right) . \qquad (5)$$

The total bending angle along the entire ray is hence obtained by finally computing the angle between initial direction at GPS and terminal direction at LEO,

$$\alpha_{total} = \arccos\left(\hat{r}_G \cdot \hat{r}_L\right) . \qquad (6)$$

Furthermore and importantly, the accumulated bending angle RIE, $\delta\alpha_{RIE}$, can be estimated (after linearly interpolating in the along-ray distance coordinate to the ray points $i$ of reference bending angle obtained without ionosphere) by subtracting the reference bending angle $\alpha_r$ from the





ionosphere-corrected bending angle $\alpha_c$ (with the latter obtained by the standard dual-frequency correction of bending angles; e.g., Liu et al., 2015; Eq. 3 therein),

$$\delta\alpha_{\text{RIE}(i)} = \alpha_{c(i)} - \alpha_{r(i)}. \tag{7}$$

As a complement to these along-ray accumulated quantities also the local bending angles and bending angle RIEs caused by individual ray tracer steps can be readily co-computed, from differencing the values between adjacent points $i+1$ and $i$,

$$\alpha_i^{\text{step}} = \alpha_{i+1} - \alpha_i, \tag{8}$$

$$\delta\alpha_{\text{RIE}(i)}^{\text{step}} = \delta\alpha_{\text{RIE}(i+1)} - \delta\alpha_{\text{RIE}(i)}. \tag{9}$$

## 3 Results and discussion

Figures 4 to 7 sequentially illustrate for the three representative RO events (Occ.25, Occ.20, Occ.530) the along-ray behavior of the key variables atmospheric and ionospheric refractivity (Eq. 4), L1 and L2 delta impact parameter (Eq. 3), L1 and L2 accumulated bending angle (Eq. 5), and ionosphere-corrected Lc bending angle and bending angle RIE (Eq. 7), respectively. This is done in form of imaging these variables for the three RO events in the along-ray distance versus impact height coordinate system (panels a and b of Figs. 4-7; ±3500 km along-ray distance about ray tangent points, impact height range 20 to 80 km) and in form of depicting the along-ray behavior of the two exceptional events along representative impact heights (80 km, 50 km, 30 km; panels c-f in Fig. 4 and panels c-d in Figs. 5-7).

In order to enable close inspection of the critical role of ionospheric symmetries, each of the Figs. 4-7 directly intercompares the non-spherical and spherical symmetry conditions. In terms of solar activity only the results for the medium solar activity level (F10.7 = 140) are illustrated, since we found that the influence of solar activity (which mainly drives the ionization level in the NeUoG model) is primarily to steer the magnitude of the effects (see Liu et al., 2013; 2015). The typical along-ray characteristics are therefore reasonably well represented by just illustrating the medium solar activity case.



Figure 4 shows the atmospheric and ionospheric refractivities and underpins that the along-ray differences of inbound ionosphere (from the GPS) and outbound ionosphere (towards the LEO) refractivities can be substantial. For example, in case of the Occ.20 event these refractivities differ by more than a factor of 2 near the ionospheric F layer maximum where the refractivities are largest (e.g.,

L2 refractivity near 10 NU on inbound while reaching more than 20 NU on outbound). As expected, the atmospheric refractivity starts to exceed 1 NU only below about 35 km and of course it exhibits no frequency dependence. It is thus essential to have a reliable first-order and higher-order ionospheric correction to strongly mitigate the ionospheric effects that appear prominent down to the lower stratosphere.

Figure 5 shows the L1 and L2 delta impact parameters, which first of all verifies the reliability of the numerical ray tracing estimates, since the spherically symmetric ionosphere conditions indeed lead to along-ray impact parameter changes of within 1 m only. This confirms that under such spherical symmetry conditions the bending angle retrievals (e.g., Schwarz et al., 2017) will be highly accurate, including for the impact height that is decisive for enabling accurate vertical geolocation

(Scherllin-Pirscher et al., 2017). Under non-spherical symmetry ionosphere conditions, the Occ.20 event with the largest asymmetry of the example events shows that along-ray L1 and L2 impact parameter variations of more than 10 m to 20 m are well possible and are generally found negative (relative to the initial impact parameter at the GPS transmitter). Lc bending angle retrievals with their intrinsic spherical symmetry assumption should thus receive higher-order ionospheric correction to

mitigate such possible impacts.

Figure 6 depicts the accumulated L1 and L2 bending angles, which highlight the significant along-ray modulations that the bending angle receives due to the ionospheric influences relative to the atmospheric bending angle, in particular above about 35 km in the upper stratosphere and mesosphere where the neutral atmospheric bending angle is rather small. Below 35 km the dominating influence of



the atmosphere in the vicinity of the tangent point location becomes prominently visible, in line with the exponential increase of atmospheric refractivity (Fig. 4) and hence atmospheric bending down into the lower stratosphere. Nevertheless even at 30 km the ionospheric contribution is still well visible, which underscores that an accurate ionospheric correction with minimized residual error will be vital.

Figure 7a shows the ionosphere-corrected Lc bending angle and indicates, compared to Figs. 6a and 6b, that the standard linear dual-frequency correction of bending angles basically does a very effective job in eliminating the ionospheric bending angle contributions. The Lc bending angle images look visually very clean and highly dominated by just the atmospheric accumulated bending angle accruing at all heights around the tangent point location. Directly inspecting the bending angle RIE, finally, shows that

the along-ray behavior and accumulated magnitude of the higher-order RIE left by the linear correction significantly depends in particular on asymmetry conditions. Also technical ray tracer effects are visible as intermittent spiky behavior, since the RIEs are at the sub-μrad to μrad magnitude level only, which is a challenge for the ionospheric model smoothness as discussed in Sect. 2.1.

The Occ.530 event under non-spherical symmetry appears to accumulate the highest RIEs of near 2 to 4

μrad at the LEO, while the spherical symmetry cases both accumulate RIEs up to around 0.5 to 1 μrad or less only. This is in line with Fig. 2, which shows for the majority of the 26 exceptional events the dominance of asymmetry effects in driving RIE magnitude. Also, as shown by Figs. 7c and 7d (and found for other RO events inspected but not separately shown), the mesospheric RIEs above about 50 km generally appear to be higher than the upper stratospheric ones from 50 km downwards. This is in

line with findings of Syndergaard (2000) and likely driven by the increased closeness of the tangent point height to the ionospheric E layer peaking near 105 km, which makes the Lc bending angle more vulnerable to higher-order RIEs.

Figures. 7c and 7d (and along-ray results for further exceptional events not separately shown) also clearly indicate the mixing-in of technical ray tracer effects in our simulations. These render it more



difficult to rigorously quantify the (smooth) physical RIE effects from the large-scale ionospheric model structure since, despite the reasonable smoothing applied, the spiky components may somewhat perturb also the smooth accumulated results. In future we will therefore aim to further improve the simulation setup to fully isolate the technical from the physics-based propagation effects. For now we found clear

evidence, nevertheless, that currently both technical effects and cases with physically high RIEs from ionospheric asymmetries play major roles in explaining the anomalous behavior of the exceptional RO events.

## 4 Summary and conclusions

Previous theoretical and simulation studies as well as empirical studies that we surveyed in the

introductory section of this study have characterized and quantified higher-order residual ionospheric errors (RIEs) in bending angles by analyses of individual events as well as ensembles of events. The statistical results showed that the mean bending angle RIE biases are predominantly negative, typically at the 0.03 to 0.1 μrad level, and that these biases may lead to systematic errors in stratospheric climatologies built from retrieved profiles. The RIE standard deviations are typically at the 0.1 to 0.5

μrad level and they have a clear tendency to increase with increasing solar activity, i.e., with increasing ionization level (electron density) in the ionosphere.

In our previous Liu et al. (2015) study we had contributed to these findings but were left with 26 exceptional RO events with very high RIEs, at the 1 to 10 μrad standard deviation level, in the context of about 700 standard events with low-noise RIEs within 0.5 μrad standard deviation. In this study we

therefore took focus on these 26 exceptional events and, by way of detailed along-ray analyses of ray tracing simulations over the stratosphere and mesosphere, inspected the causes of anomalously high RIEs. The goal at the same time was to deepen quantitative insight into how RIEs accumulate during





signal propagation, along with accumulation of the total atmospheric bending angles that are the desired RO observables.

From the results of these analyses we conclude with the following main findings on the causes for the exceptional RO events:

1.) asymmetric ionospheric conditions play the primary role for anomalously high RIEs, more than ionization levels driven by solar activity and possible geographic location dependencies that seemed to be present from salient geographic clustering of the majority of exceptional RO events in two regions (European Asian region and Indian Ocean region);

2.) the fact that no obvious physics-based explanation was found for the geographic clustering and the intermittent spiky behaviour found in simulated RIEs indicates that part of the anomalous RIEs of the exceptional RO events were caused by the technical challenge of ray tracing at millimetric excess phase accuracy through the 3D ionospheric model NeUoG that is not perfectly smooth everywhere in its electron density field derivatives;

3.) the detailed along-ray analyses of atmospheric and ionospheric refractivities, impact parameter changes, bending angles and RIEs also revealed that along-ray L1 and L2 impact parameter variations of more than 10 m to 20 m are well possible due to ionospheric asymmetries and are generally found negative (relative to the initial impact parameter at the GPS transmitter). Standard bending angle retrievals with their intrinsic spherical symmetry assumption should thus receive higher-order ionospheric correction to mitigate such impacts:

4.) the mesospheric RIEs above about 50 km generally appear to be higher than the upper stratospheric ones from 50 km downwards. This is in line with findings of Syndergaard (2000) and likely driven by the increased closeness of the tangent point height to the ionospheric E layer peaking near 105 km, which makes the standard ionosphere-corrected bending angles more vulnerable to higher-order RIEs.



Overall this study of exceptional RO events with anomalous RIEs in our end-to-end simulations indicated that the main causes are a combination of physics-based effects, in particular ionospheric asymmetries, and of technical ray tracer effects due to occasionally imperfect smoothness of modeling ionospheric refractivity field derviatives. This makes it more difficult to rigorously quantify the physics-based RIE effects from the large-scale ionospheric model structure since the intermittent spiky nature of the technical effects may somewhat perturb also the smooth accumulated results.

In future we will therefore aim to further improve our along-ray simulation and analysis system to fully isolate the technical from the physics-based propagation effects. For now we found clear evidence, nevertheless, that currently both technical effects and cases with physically high RIEs from ionospheric asymmetries play major roles in explaining the anomalous behavior of the exceptional RO events. The detailed along-ray modeling system will also be valuable beyond this work for additional GNSS RO signal propagation studies.

## Acknowledgements

This research was partially supported by the National Natural Science Foundation of China (Grant Nos. 41405039, 41775034, 41405040, 41505030 and 41606206) and the FengYun-3 (FY-3) Global Navigation Satellite System Occultation Sounder (GNOS) development and manufacture project led by NSSC, CAS. The research at WEGC/Univ. of Graz was supported by the European Space Agency (ESA) projects OPSGRAS and MMValRO and the Austrian Research Promotion Agency (FFG) project OPSCLIMPROP (ASAP-9 Project No. 840070). We acknowledge J. Fritzer (WEGC) for his support in EGOPS software developments valuable for this study. The research at SPACE/RMIT University was supported by the Australian Research Council (ARC) (LP0883288), the Australian Antarctic Division (Project No. 4159) and the CAS/SAFEA International Partnership Program for Creative Research Teams (Grant No. KZZD-EW-TZ-05). The ECMWF (Reading, UK) is thanked for access to their



archived analysis and forecast data (more information available at http://www.ecmwf.int/en/ forecasts/datasets). The software code used for this study does not belong to the public domain and cannot be distributed. To access the relevant result files of this study, please contact the corresponding author.

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



**Table 1.** Parameters of the three representative RO events used for detailed inspection. Azimuth of the RO event plane is defined relative to North, counting over West.

| Event.Id | Latitude | Longitude | Azimuth | Local time | Solar activity |
|---|---|---|---|---|---|
| Occ.530 | 55.8 °S | 61.8 °E | 167.2 ° | 21:38 LT | F10.7=140 |
| Occ.20 | 43.3 °N | 36.5 °W | 154.1 ° | 22:34 LT | F10.7=140 |
| Occ.25 | 81.1 °N | 5.4 °W | 94.1 ° | 01:09 LT | F10.7=140 |





## Figure captions

**Figure 1.** Radio occultation geometry between GNSS transmitter and low Earth orbit (LEO) receiver satellites, schematically illustrating the separate L1 and L2 signal ray paths and the ionosphere-corrected (Lc) ray path through the atmosphere-ionosphere system. Key quantities additionally indicated are the (total accumulated) bending angle $\alpha$, the (spherically symmetric) ray impact parameter $a$, and the radius $r$ from the Earth's center of curvature to the tangent point of the Lc signal path (modified from Liu et al., 2015).

**Figure 2.** Distribution of the mean tangent point locations of the 723 RO events simulated by Liu et al. (2015) for 15 July 2008 **(a)**, including 697 events with standard RIE (small-white triangles) and 26 events with exceptional RIE (red triangles; upward-pointing, rising events; downward-pointing, setting events). The latter 26 events mainly reside in the European Asian Cluster (EAC; magenta box) and Indian Ocean Cluster (IOC; green box), respectively. The background color map illustrates the vertically-integrated Total Electron Content (vTEC) of the NeUoG ionospheric model for medium solar activity (for 12:00 UTC of $15^{th}$ July; F10.7 = 140; shown in TEC units, 1 TECU = $10^{16}$ electrons $m^{-2}$). The bottom panels depict the RIEs for perfect observing system (op) with no observational errors and non-spherical (opns) **(b)** as well as spherically symmetric (opss) **(c)** ionospheric conditions. They show the bending angle RIE bias (symbols) and standard deviation (error bars) estimates for the 30-80 km range for low (F10.7 = 70, green), medium (F10.7 = 140, blue), and high (F10.7 = 210, red) solar activity, for each of the 26 exceptional events (ordered by clusters, with those not falling into EAC and IOC marked as OTHERS), with each one identified by its chronological RO event number of the day.

**Figure 3.** Illustration of the three example events chosen for detailed inspection (Occ.530, red; Occ.20, green; Occ.25, blue), showing their excess phase profiles **(a)**, excess phase RIE profiles **(b)**, bending angle profiles **(c)**, and bending angle RIE profiles **(d)**, respectively, over the impact height range 40 to 80 km for medium solar activity (F10.7=140) and non-spherical symmetry (ns) ionosphere conditions. The excess phase and bending angle profiles are shown for both GPS frequencies L1 (dashed) and L2 (dashed-dotted) as well as after standard first-order ionospheric correction (subscript c; solid).

**Figure 4.** Images of the atmospheric and ionospheric refractivity (Eq. 4) in the along-ray distance (relative to tangent point) versus impact height coordinate system for non-spherical symmetry **(a)** and spherical symmetry **(b)** ionospheric conditions, for medium solar activity (F10.7=140) and for the GPS frequencies L1 (left sub-panels) and L2 (right sub-panels) for the three representative events (Occ.25, top sub-panels; Occ.20, middle sub-panels, Occ.530, bottom sub-panels). The narrow black stripe visible in the images near the right margin (3500 km along-ray distance) is space above the LEO orbit height (reached at around 3250 km). The two bottom rows depict the corresponding along-ray behavior of the atmospheric **(c, d)** and ionospheric **(e, f)** refractivities at three representative impact heights (red, 80 km; green, 50 km; blue, 30 km) for non-spherical (left; c and e) and spherical (right; d and f) ionospheric conditions, for the Occ.20 (left sub-panel in c-f) and Occ.530 (right





sub-panel in c-f) event. The sub-panel titles (green in a-b panels, black on top of c-f panels) identify the individual cases by a concise acronym; the physical units used are N units (1 NU = $10^{6} \cdot (n–1)$).

**Figure 5.** Images of the delta impact parameter (Eq. 3) in the along-ray distance (relative to tangent point) versus impact height coordinate system for non-spherical symmetry **(a)** and spherical symmetry **(b)** ionospheric conditions, for medium solar activity (F10.7=140) and for the GPS frequencies L1 (left sub-panels) and L2 (right sub-panels) for the three representative events (Occ.25, top sub-panels; Occ.20, middle sub-panels, Occ.530, bottom sub-panels). The narrow black stripe visible in the images near the right margin (3500 km along-ray distance) is space above the LEO orbit height (reached at around 3250 km). The bottom row depicts the corresponding along-ray behavior of the delta impact parameter at three representative impact heights (red, 80 km; green, 50 km; blue, 30 km) for non-spherical **(c)** and spherical **(d)** ionospheric conditions, for the Occ.20 (left sub-panels) and Occ.530 (right sub-panels) event. The sub-panel titles (red in a-b subpanels, black on top of c-d subpanels) identify the individual cases by a concise acronym.

**Figure 6.** Images of the accumulated bending angle (Eq. 5) in the along-ray distance versus impact height coordinate system for non-spherical symmetry **(a)** and spherical symmetry **(b)** ionospheric conditions, and the corresponding along-ray behavior at three selected impact heights **(c, d)**. All panels are shown in the same format as for the delta impact parameter in Fig. 5 (see that caption for more details). Here the physical units are [μrad].

**Figure 7.** Images of the accumulated ionosphere-corrected Lc bending angle [μrad] **(a)** and bending angle RIE [μrad] **(b)** (Eq. 7) in the along-ray distance versus impact height coordinate system for non-spherical symmetry (left sub-panels) and spherical symmetry (right sub-panels) ionospheric conditions, for medium solar activity (F10.7=140) and the same three representative events also shown in Figs. 4-6. The bottom row **(c, d)** depicts the corresponding along-ray behavior of the accumulated bending angle RIE at three representative impact heights (red, 80 km; green, 50 km; blue, 30 km) in the same format as in Figs. 4-6. Since the raw RIE estimates (light lines in c-d, with intermittent spiky behavior) are noisy due to technical ray tracing effects from limited smoothness of the NeUoG model (Sect. 2.1), the essential behavior (heavy lines in c-d, with smooth behavior) is shown with the RIE data smoothed along-ray by a first-median-then-mean filter (using ±350 km moving median filter width, then ±150 km moving average filter width).



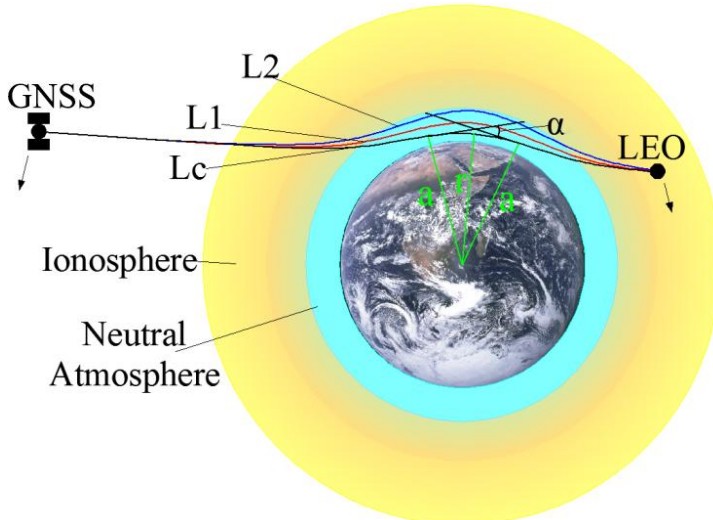

**Figure 1.** Radio occultation geometry between GNSS transmitter and low Earth orbit (LEO) receiver satellites, schematically illustrating the separate L1 and L2 signal ray paths and the ionosphere-corrected (Lc) ray path through the atmosphere-ionosphere system. Key quantities additionally indicated are the (total accumulated) bending angle $\alpha$, the (spherically symmetric) ray impact parameter $a$, and the radius $r$ from the Earth's center of curvature to the tangent point of the Lc signal path (modified from Liu et al., 2015).




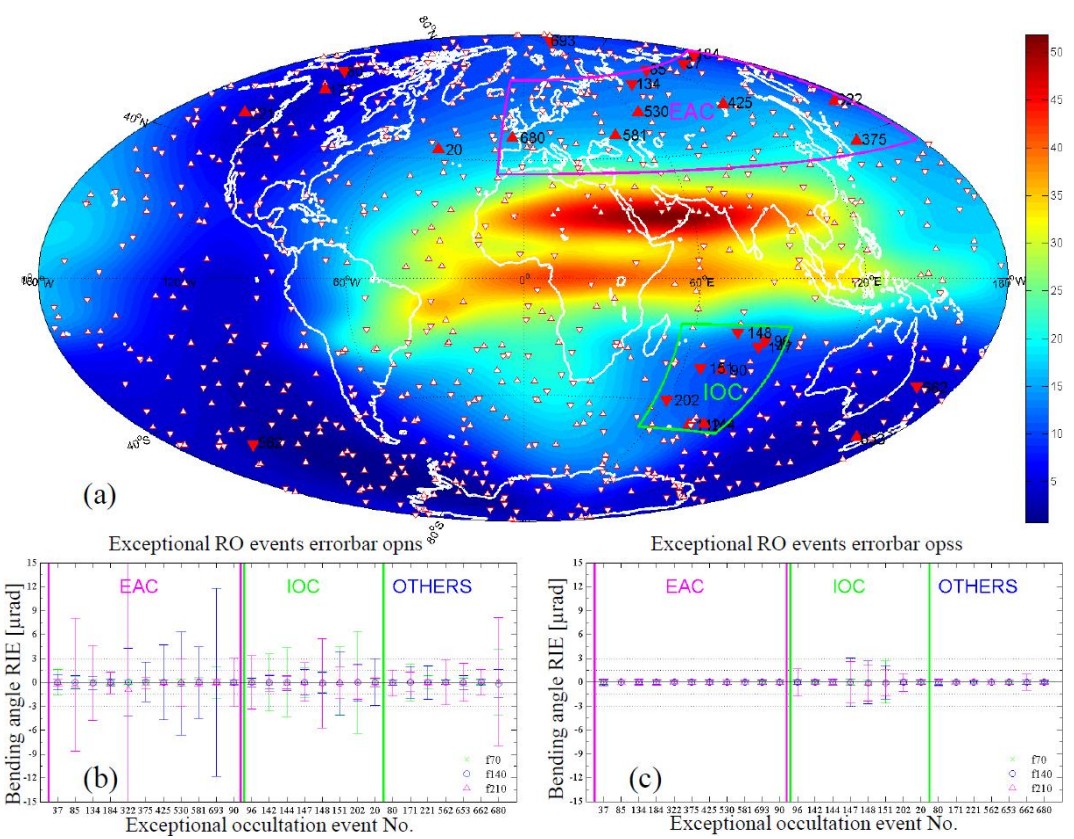

**Figure 2.** Distribution of the mean tangent point locations of the 723 RO events simulated by Liu et al. (2015) for 15 July 2008 **(a)**, including 697 events with standard RIE (small-white triangles) and 26 events with exceptional RIE (red triangles; upward-pointing, rising events; downward-pointing, setting events). The latter 26 events mainly reside in the European Asian Cluster (EAC; magenta box) and Indian Ocean Cluster (IOC; green box), respectively. The background color map illustrates the vertically-integrated Total Electron Content (vTEC) of the NeUoG ionospheric model for medium solar activity (for 12:00 UTC of 15[th] July; F10.7 = 140; shown in TEC units, 1 TECU = $10^{16}$ electrons m$^{-2}$). The bottom panels depict the RIEs for perfect observing system (op) with no observational errors and non-spherical (opns) **(b)** as well as spherically symmetric (opss) **(c)** ionospheric conditions. They show the bending angle RIE bias (symbols) and standard deviation (error bars) estimates for the 30-80 km range for low (F10.7 = 70, green), medium (F10.7 = 140, blue), and high (F10.7 = 210, red) solar activity, for each of the 26 exceptional events (ordered by clusters, with those not falling into EAC and IOC marked as OTHERS), with each one identified by its chronological RO event number of the day.




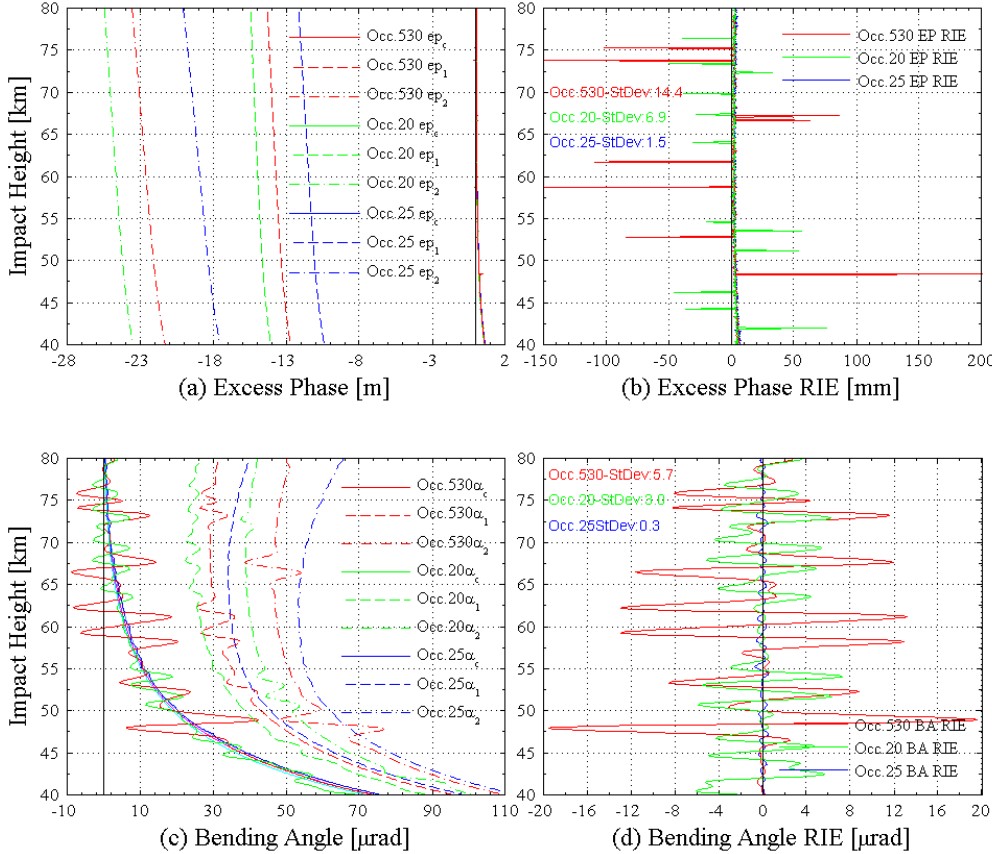

**Figure 3.** Illustration of the three example events chosen for detailed inspection (Occ.530, red; Occ.20, green; Occ.25, blue), showing their excess phase profiles **(a)**, excess phase RIE profiles **(b)**, bending angle profiles **(c)**, and bending angle RIE profiles **(d)**, respectively, over the impact height range 40 to 80 km for medium solar activity (F10.7=140) and non-spherical symmetry (ns) ionosphere conditions. The excess phase and bending angle profiles are shown for both GPS frequencies L1 (dashed) and L2 (dashed-dotted) as well as after standard first-order ionospheric correction (subscript c; solid).





**Figure 4.** Images of the atmospheric and ionospheric refractivity (Eq. 4) in the along-ray distance (relative to tangent point) versus impact height coordinate system for non-spherical symmetry **(a)** and spherical symmetry **(b)** ionospheric conditions, for medium solar activity (F10.7=140) and for the GPS frequencies L1 (left sub-panels) and L2 (right sub-panels) for the three representative events (Occ.25, top sub-panels; Occ.20, middle sub-panels, Occ.530, bottom sub-panels). The narrow black stripe visible in the images near the right margin (3500 km





along-ray distance) is space above the LEO orbit height (reached at around 3250 km). The two bottom rows depict the corresponding along-ray behavior of the atmospheric **(c, d)** and ionospheric **(e, f)** refractivities at three representative impact heights (red, 80 km; green, 50 km; blue, 30 km) for non-spherical (left; c and e) and spherical (right; d and f) ionospheric conditions, for the Occ.20 (left sub-panel in c-f) and Occ.530 (right sub-panel in c-f) event. The sub-panel titles (green in a-b panels, black on top of c-f panels) identify the individual cases by a concise acronym; the physical units used are N units (1 NU = $10^6 \cdot (n-1)$).

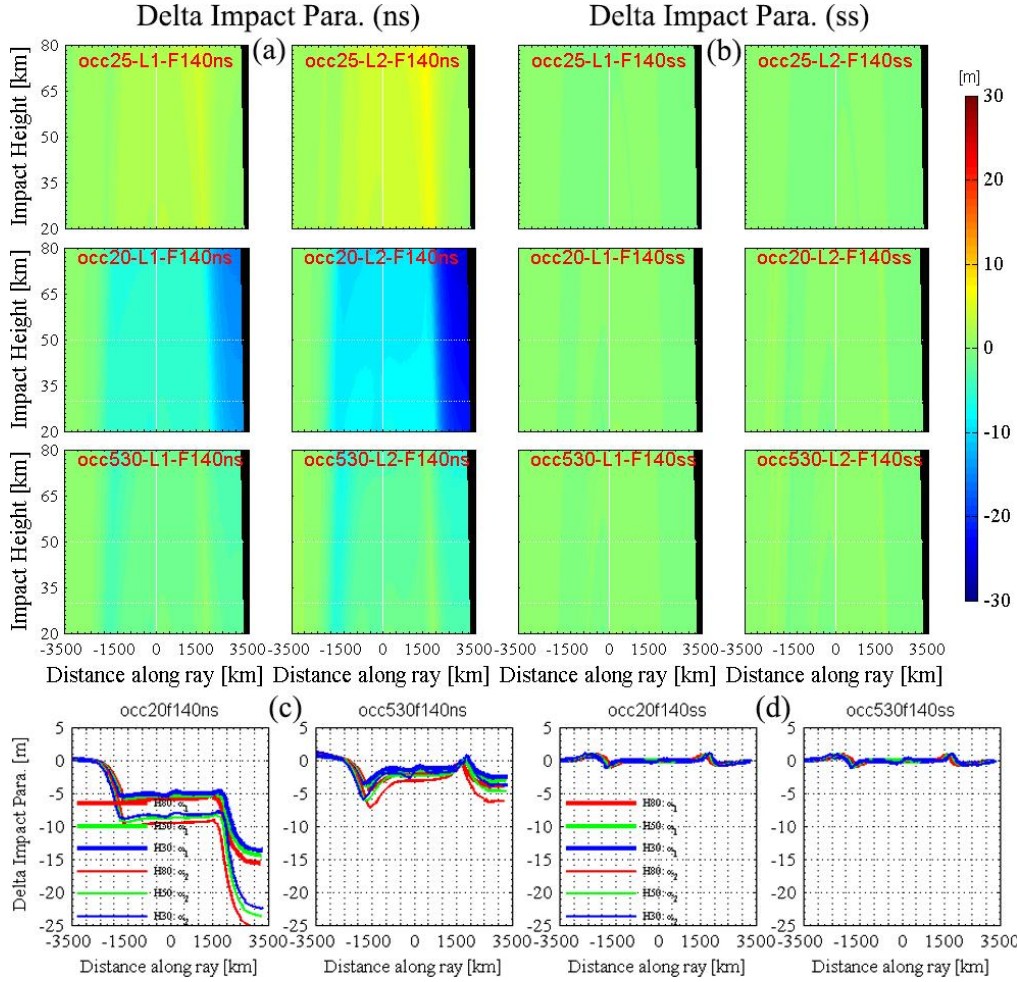

**Figure 5.** Images of the delta impact parameter (Eq. 3) in the along-ray distance (relative to tangent point) versus impact height coordinate system for non-spherical symmetry **(a)** and spherical symmetry **(b)** ionospheric conditions, for medium solar activity (F10.7=140) and for the GPS frequencies L1 (left sub-panels) and L2 (right sub-panels) for the three representative events (Occ.25, top sub-panels; Occ.20, middle sub-panels, Occ.530, bottom sub-panels). The narrow black stripe visible in the images near the right margin (3500 km along-ray distance) is space above the LEO orbit height (reached at around 3250 km). The bottom row depicts the




corresponding along-ray behavior of the delta impact parameter at three representative impact heights (red, 80 km; green, 50 km; blue, 30 km) for non-spherical **(c)** and spherical **(d)** ionospheric conditions, for the Occ.20 (left sub-panels) and Occ.530 (right sub-panels) event. The sub-panel titles (red in a-b subpanels, black on top of c-d subpanels) identify the individual cases by a concise acronym.

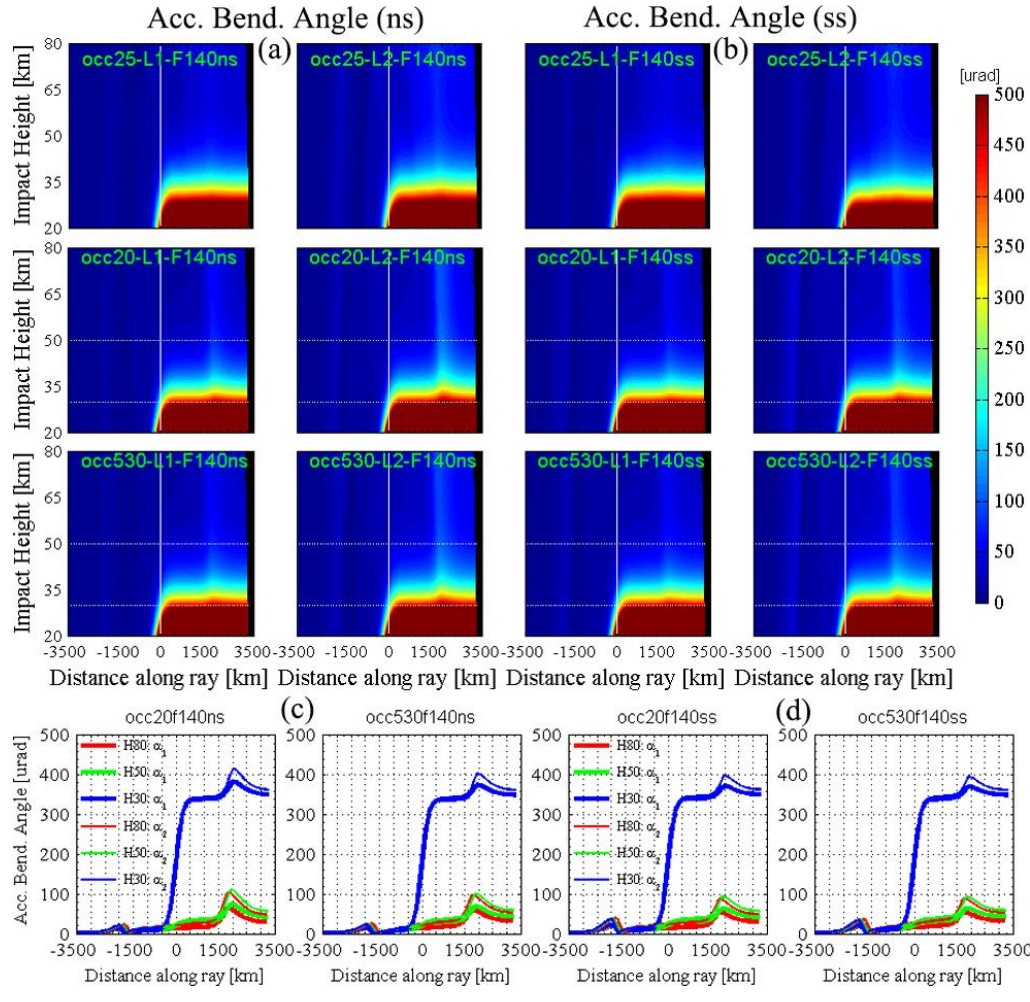

**Figure 6.** Images of the accumulated bending angle (Eq. 5) in the along-ray distance versus impact height coordinate system for non-spherical symmetry **(a)** and spherical symmetry **(b)** ionospheric conditions, and the corresponding along-ray behavior at three selected impact heights **(c, d)**. All panels are shown in the same format as for the delta impact parameter in Fig. 5 (see that caption for more details). Here the physical units are [μrad].





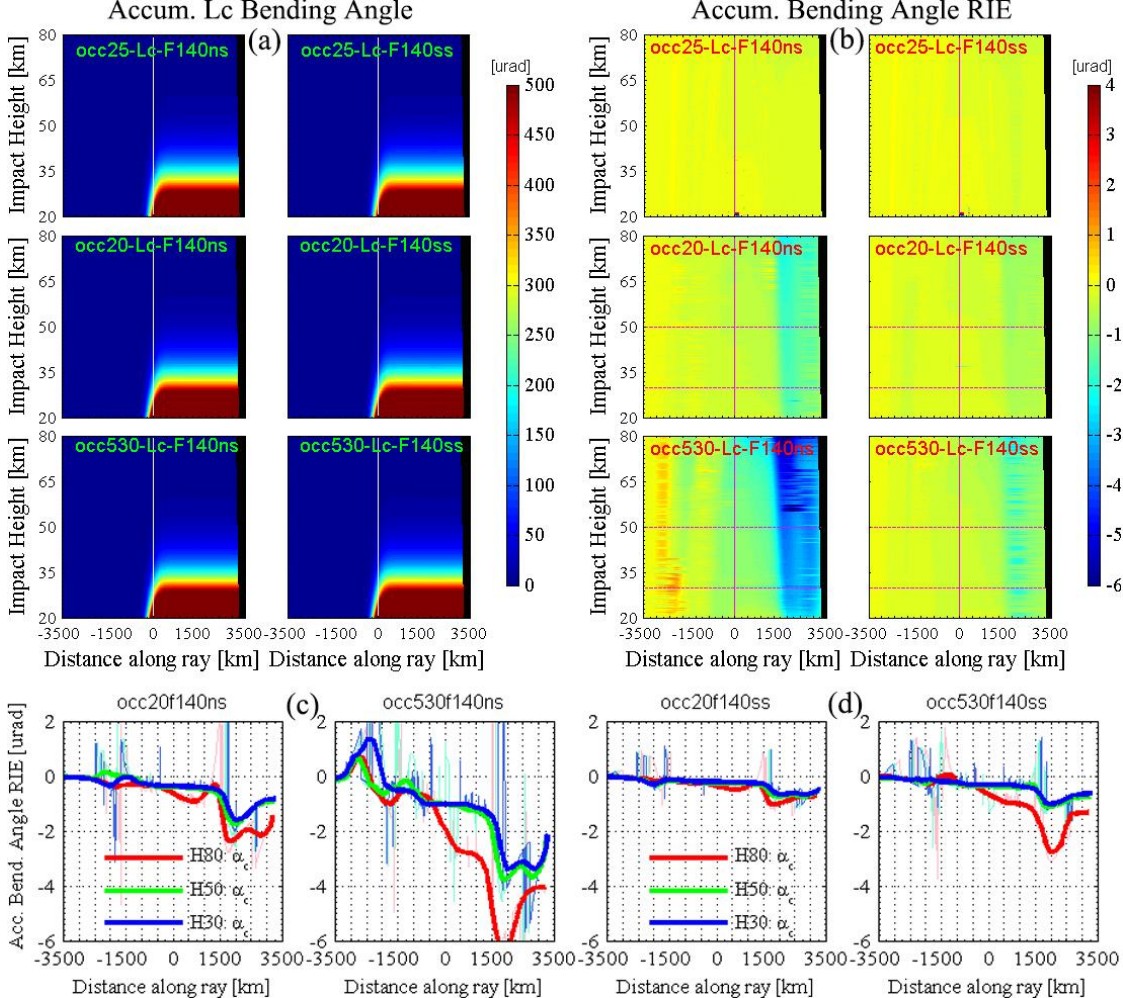

**Figure 7.** Images of the accumulated ionosphere-corrected Lc bending angle [µrad] **(a)** and bending angle RIE [µrad] **(b)** (Eq. 7) in the along-ray distance versus impact height coordinate system for non-spherical symmetry (left sub-panels) and spherical symmetry (right sub-panels) ionospheric conditions, for medium solar activity (F10.7=140) and the same three representative events also shown in Figs. 4-6. The bottom row **(c, d)** depicts the corresponding along-ray behavior of the accumulated bending angle RIE at three representative impact heights (red, 80 km; green, 50 km; blue, 30 km) in the same format as in Figs. 4-6. Since the raw RIE estimates (light lines in c-d, with intermittent spiky behavior) are noisy due to technical ray tracing effects from limited smoothness of the NeUoG model (Sect. 2.1), the essential behavior (heavy lines in c-d, with smooth behavior) is shown with the RIE data smoothed along-ray by a first-median-then-mean filter (using ±350 km moving median filter width, then ±150 km moving average filter width).