# Peer review of "Analysis of ionospheric structure influences on residual ionospheric errors in GNSS radio occultation bending angles based on ray tracing simulations"

_Atmospheric Measurement Techniques, 2017_

## Referee Comment (RC1) · Anonymous Referee #1 · 25 Oct 2017

This paper presents a detailed analysis of residual ionospheric errors (RIE) that have been found in a simulation study of radio occultation measurements. It is, in effect, an extension of [Liu et al., 2015], where the simulation was first reported. In that paper the data with large RIE was excluded from the analysis. In this paper, the large RIE data is re-analysed in order to assess, in detail, how the errors accrue along the raypath. This attention to detail is commendable and provides useful insight into the measurement. The paper is clearly worthy of publication in AMT.

One concern is that the conclusions may not be fully supported by the text. In partic-

ular, the authors make the point that the large RIEs can be produced by ionospheric asymmetry or by technical ray-tracing errors that probably arise from discontinuities in NeUoG. The role of ionospheric asymmetry to emphasized by showing that the errors largely disappear when a spherically symmetric ionosphere is used, However, it seems likely that any NeUoG discontinuities will also be removed in the spherically symmetric case; i.e. the two issues cannot by separated by this test. It would be more persuasive if it could be demonstrated that the rest of the dataset (i.e. those with reasonable RIEs) did not exhibit ionospheric asymmetries. Through since no physical reason is presented for the large RIEs to occur in the geographic areas where they are most prevalent, it seems this is unlikely. If ionospheric asymmetries do occur in the other data, the conclusion may be that the large RIEs are caused by the ray-trace problems alone, or by a combination of both the asymmetry and the ray-trace.

Other issues:

The work of [Danzer et al., 2015] is referenced. In that paper the analysis was limited by "high noise of the simulated bending-angle profiles at mid- to high latitudes". Is this the same problem ray-trace? If so, it is probably worth mentioning it.

Page 4, line 17. This sentence has become confused and a rogue full stop is present.

Refs

Danzer, J., S. B. Healy, and I. D. Culverwell (2015), A simulation study with a new residual ionospheric error model for GPS radio occultation climatologies, Atmos. Meas. Tech, 8, 3395–3404, doi:10.5194/amt-8-3395-2015.

Liu, C. L., G. Kirchengast, K. Zhang, R. Norman, Y. Li, S. C. Zhang, J. Fritzer, M. Schwaerz, S. Q. Wu, and Z. X. Tan (2015), Quantifying residual ionospheric errors in GNSS radio occultation bending angles based on ensembles of profiles from end-to-end simulations, Atmos. Meas. Tech., 8(7), 2999–3019, doi:10.5194/amt-8-2999-2015.

---

## Referee Comment (RC2) · Anonymous Referee #2 · 9 Dec 2017

Summary: The authors perform a study of residual ionospheric error (RIE) in radio occultation simulations. They build on a past study to focus on 26 events with anomalously large RIEs that appear at the 1 to 10 micro-rad level. The authors analyze how the residual error changes along the raypath. Asymmetric ionospheric conditions are found to play a primary role in the largest RIEs, more than the ionization level driven by solar activity and other factors. Another factor is technical ray tracer effects due to occasionally imperfect smoothness in ionospheric refractivity model derivatives, but not the main one. Mesospheric RIEs are generally found to be higher than in the upper

stratosphere, likely due to proximity of the tangent point to the source region of largest asymmetry, the ionospheric E layer.

Review summary:

The authors successfully build on prior work to create a useful and interesting article on what causes the largest residual ionospheric errors. As far as I know, there is no other work of this kind, and it provides useful knowledge on the effects of spherical asymmetry. Certain aspects of the presentation should be improved before publication. These are described in this section and in the detailed comments.

It would be very useful to the conclusions if the authors could show that the regions of largest RIE are also regions of large asymmetry. Perhaps they are regions of large ray-tracer error also, due to the NeUoG algorithm? This is unlikely, but it is also not clear why large ionospheric asymmetry is found in the regions highlighted by this study. The authors should consider this point more carefully. What would be useful is a metric of asymmetry for each occultation, and then showing that for nearly all the occultations of small to moderate RIE, the asymmetry is smaller than for nearly all of the occultations of large RIE studied here. Without this additional information one runs the risk of ascribing asymmetry as the cause of large RIE by coincidence. It might be the case that the large RIE differences between symmetric and asymmetric conditions is correlated with some other reason the RIEs are so large in the highlighted regions. Are the regions of large RIE also the regions where ionospheric asymmetries are largest, based on fundamental ionospheric considerations independent of the raytracing?

The authors should consider citing a recent publication that is related to their work: Coleman, C. J., and B. Forte (2017), On the residual ionospheric error in radio occultation measurements, Radio Sci., 47(3), 653–20, doi:10.1002/2016RS006239.

In the spirit of remarks made by co-author Kirchengast at IROWG-6 on referencing unpublished work, the authors might consider referencing the presentation by Mannucci et al., at OPAC-2010, on the role of ionospheric asymmetry in the magnitude of the RIE.

If the authors of this paper believe that Mannucci et al.'s presentation is not sufficiently relevant, no reference is needed.

The authors tend not to emphasize ionospheric asymmetry in their 2015 published study, whereas it factors into this study in a major way. A remark or two on why this study found asymmetry as important given the prior 2015 study is recommended.

Detailed Comments:

Page 4, Line 14: The term "quasi-realistic" is somewhat vague. It is understood that simulations are approximations of reality, so what is meant by this phrase?

Page 4, Line 20: I believe unpublished work by Mannucci et al. (OPAC 2010) contained an initial analysis of the role of ionospheric asymmetry. Certainly, the present work is more comprehensive than was presented then.

Page 6, Line 16: "clustered"

Page 6, Line 17: How is standard deviation defined for the bottom panels of Figure 2?

P6, L19: how is spherical symmetry (SS) achieved in the model? Please provide a brief description.

P7, L6: what is meant by "geographical dependencies" as separated from SS conditions or similar? Pure geography (latitude and longitude) would not be a factor, in general.

P11, L19: are out-of-occultation-plane refractive gradients ignored or included? Please clarify.

P17, L5: to bolster this conclusion and its applicability to real observations, it would be useful to see a metric of ionospheric asymmetry in other regions, to show that in the regions highlighted here, ionospheric asymmetry is largest. If ionospheric asymmetries are large in other regions also, but RIE is not so large, then how robust is this conclusion?

---

## Author Comment (AC1) · 28 Feb 2018

Manuscript doi:10.5194/amt-2017-242, 2017 Manuscript Title: Analysis of ionospheric structure influences on residual ionospheric errors in GNSS radio occultation bending angles based on ray tracing simulations Authors: Congliang Liu, Gottfried Kirchengast, Yueqiang Sun, Kefei Zhang, Robert Norman, Marc Schwaerz, Weihua Bai, Qifei Du, and Ying Li

(PS: A more user friendly pdf version of this response has been attached in supple-

**ment.)**

We thank the referee very much for the constructive comments and recommendations and for the overall positive rating that this is considered a useful paper clearly worthy of publication. We thoroughly considered all comments and carefully revised the manuscript accounting for most of them. In addition, we carefully complemented these revisions with a couple of further improvements throughout the manuscript text in the spirit of the comments. Please find below our point-by-point response (in form of italicized, blue text) to the referees' comments (in form of upright, black text), inserted below each comment. Line numbers used in our responses refer to the original AMT Discussions paper and text updates in the revised manuscript are quoted below with yellow highlighting.

Response to Anonymous Referee #1's Comments

1. General Comments This paper presents a detailed analysis of residual ionospheric errors (RIE) that have been found in a simulation study of radio occultation measurements. It is, in effect, an extension of [Liu et al., 2015], where the simulation was first reported. In that paper the data with large RIE was excluded from the analysis. In this paper, the large RIE data is re-analysed in order to assess, in detail, how the errors accrue along the raypath. This attention to detail is commendable and provides useful insight into the measurement. The paper is clearly worthy of publication in AMT. Thank you.

One concern is that the conclusions may not be fully supported by the text. In particular, the authors make the point that the large RIEs can be produced by ionospheric asymmetry or by technical ray-tracing errors that probably arise from discontinuities in NeUoG. The role of ionospheric asymmetry to emphasized by showing that the errors largely disappear when a spherically symmetric ionosphere is used, However, it seems likely that any NeUoG discontinuities will also be removed in the spherically symmetric case; i.e. the two issues cannot by separated by this test. It would be more persuasive if it could be demonstrated that the rest of the dataset (i.e. those with reasonable RIEs) did not exhibit ionospheric asymmetries. Through since no physical reason is presented for the large RIEs to occur in the geographic areas where they are most prevalent, it seems this is unlikely. If ionospheric asymmetries do occur in the other data, the conclusion may be that the large RIEs are caused by the ray-trace problems alone, or by a combination of both the asymmetry and the ray-trace. Thank you for this important comment; we also got similar questions from the second referee. Based on these comments we carefully re-assessed the 26 cases in terms of their asymmetry, also in the context of the other dataset with the reasonable RIEs, and found re-confirmed that the physical asymmetry and the technical effects inevitably mix up as long as we do not have an advanced ray tracing based on rigorously smooth 3D ionospheric modeling that reliably keeps the technical effects negligible. (Despite efforts, including talking to other relevant ionospheric experts such as P.Straus/Aerospace Corp. and Stig Syndergaard/DMI, we could not get to such a ray-tracing-using-rigorouslysmooth-iono.modeling solution yet.) We therefore toned down the related discussion a bit now, at several places in the text where found better, including toning down also the conclusion on the role of iono.asymmetry. We re-checked the abstract first and think in this one we got the right tone already, including that in the last sentence we clearly point to the needed further improvement. In the conclusions we changed, on p. 17, lines 5-6, from "asymmetric ionospheric conditions play the primary role for anomalously high RIEs," to "strengthening previous results by Mannucci et al. (2010, 2011) we find that asymmetric ionospheric conditions play an important role for anomalously high RIEs," Otherwise we think it looks adequate, again clearly making the point at the end of the section towards the needed further improvement. In the remainder of the text we changed as follows, at places where we deemed it relevant: on p. 6, lines 20-21, we rephrased from "main driver of anomalously high RIEs are asymmetric ionospheric conditions as only few events" to "main driver of anomalously high RIEs are asymmetric ionospheric conditions and possibly residual error effects from ray tracing through the 3D ionosphere, since only few events"; on p. 7, line 9, we changed from "some

СЗ

perturbations may also come in from" to "some perturbations also come in from"; on p. 15, line 17, we replaced "dominance of asymmetry effects in driving" by "dominance of asymmetry and 3D ray tracing effects in driving the"; and on p. 16, line 6, we updated from "play major roles" to "play important roles".

2. Specific Comments Other issues: The work of [Danzer et al., 2015] is referenced. In that paper the analysis was limited by "high noise of the simulated bending-angle profiles at mid- to high latitudes". Is this the same problem ray-trace? If so, it is probably worth mentioning it. Yes, was the same kind of limitation. Ok, we added on p. 9, line 13, a sentence which mentions this: "Danzer et al. (2015) noted that their analysis was somewhat limited by high noise of the simulated bending angle profiles at mid- to high latitudes, which partly reflected the degrading impact of technical ray tracer effects that we also encounter and more explicitly address in this study."

Page 4, line 17. This sentence has become confused and a rogue full stop is present. We agree this is confusing currently; we thus improved the current p. 4, line 16-19, text part to: "...the RIE biases have a clear negative tendency and a magnitude increasing with solar activity as well as are affected by deviations from ionospheric spherical symmetry (Mannucci et al., 2010) where increasing asymmetries also tend to increase the biases."

Refs Danzer, J., S. B. Healy, and I. D. Culverwell (2015), A simulation study with a new residual ionospheric error model for GPS radio occultation climatologies, Atmos. Meas. Tech, 8, 3395–3404, doi:10.5194/amt-8-3395-2015. Liu, C. L., G. Kirchengast, K. Zhang, R. Norman, Y. Li, S. C. Zhang, J. Fritzer, M. Schwaerz, S. Q. Wu, and Z. X. Tan (2015), Quantifying residual ionospheric errors in GNSS radio occultation bending angles based on ensembles of profiles from end-to-end simulations, Atmos. Meas. Tech., 8(7), 2999–3019, doi:10.5194/amt-8-2999-2015.

Please also note the supplement to this comment: https://www.atmos-meas-tech-discuss.net/amt-2017-242/amt-2017-242-AC1-

**supplement.pdf**

---

## Author Comment (AC2) · 28 Feb 2018

Manuscript doi:10.5194/amt-2017-242, 2017 Manuscript Title: Analysis of ionospheric structure influences on residual ionospheric errors in GNSS radio occultation bending angles based on ray tracing simulations Authors: Congliang Liu, Gottfried Kirchengast, Yueqiang Sun, Kefei Zhang, Robert Norman, Marc Schwaerz, Weihua Bai, Qifei Du, and Ying Li

(PS: A more user friendly pdf version of this response has been attached in supple-

**ment.)**

We thank the referee very much for the constructive comments and recommendations and for the overall positive rating that this is a useful and interesting scientific paper. We thoroughly considered all comments and carefully revised the manuscript accounting for most of them. In addition, we carefully complemented these revisions with a couple of further improvements throughout the manuscript text in the spirit of the comments. Please find below our point-by-point response (in form of italicized, blue text) to the referees' comments (in form of upright, black text), inserted below each comment. Line numbers used in our responses refer to the original AMT Discussions paper and text updates in the revised manuscript are quoted below with yellow highlighting.

Response to Anonymous Referee #2's Comments

**1. General Comments**

Summary: The authors perform a study of residual ionospheric error (RIE) in radio occultation simulations. They build on a past study to focus on 26 events with anomalously large RIEs that appear at the 1 to 10 micro-rad level. The authors analyze how the residual error changes along the raypath. Asymmetric ionospheric conditions are found to play a primary role in the largest RIEs, more than the ionization level driven by solar activity and other factors. Another factor is technical ray tracer effects due to occasionally imperfect smoothness in ionospheric refractivity model derivatives, but not the main one. Mesospheric RIEs are generally found to be higher than in the upper stratosphere, likely due to proximity of the tangent point to the source region of largest asymmetry, the ionospheric E layer. Review summary: The authors successfully build on prior work to create a useful and interesting article on what causes the largest residual ionospheric errors. As far as I know, there is no other work of this kind, and it provides useful knowledge on the effects of spherical asymmetry. Certain aspects of the presentation should be improved before publication. These are described in this section and in the detailed comments. Thank you for this summary; we have revised

the paper in line with our responses below.

It would be very useful to the conclusions if the authors could show that the regions of largest RIE are also regions of large asymmetry. Perhaps they are regions of large raytracer error also, due to the NeUoG algorithm? This is unlikely, but it is also not clear why large ionospheric asymmetry is found in the regions highlighted by this study. The authors should consider this point more carefully. What would be useful is a metric of asymmetry for each occultation, and then showing that for nearly all the occultations of small to moderate RIE, the asymmetry is smaller than for nearly all of the occultations of large RIE studied here. Without this additional information one runs the risk of ascribing asymmetry as the cause of large RIE by coincidence. It might be the case that the large RIE differences between symmetric and asymmetric conditions is correlated with some other reason the RIEs are so large in the highlighted regions. Are the regions of large RIE also the regions where ionospheric asymmetries are largest, based on fundamental ionospheric considerations independent of the raytracing? Thank you for these important considerations and concerns related to adequate interpretation of the asymmetries from the results of this study; we also got a similar comment from the first referee. Based on these, we carefully re-assessed the 26 cases in terms of their asymmetry, also in the context of the other dataset with the reasonable RIEs, and found re-confirmed that the physical asymmetry and the technical effects inevitably mix up as long as we do not have an advanced ray tracing based on rigorously smooth 3D ionospheric modeling that reliably keeps the technical effects negligible. Despite efforts (including talking to other relevant ionospheric experts such as P.Straus/Aerospace Corp. and Stig Syndergaard/DMI) such a ray tracing using rigorously smooth 3D iono.modeling is still beyond the scope of this study. We therefore toned down the related discussion a bit now, at several places in the text where found better, including toning down also the conclusion on the role of iono.asymmetry. We re-checked the abstract first and think in this one we got the right tone already, including that in the last sentence we clearly point to the needed further improvement. In the conclusions we changed, on p. 17, lines 5-6, from "asymmetric ionospheric con-

СЗ

ditions play the primary role for anomalously high RIEs," to "strengthening previous results by Mannucci et al. (2010, 2011) we find that asymmetric ionospheric conditions play an important role for anomalously high RIEs," Otherwise we think it looks adequate, again clearly making the point at the end of the section towards the needed further improvement. In the remainder of the text we changed as follows, at places where we deemed it relevant: on p. 6, lines 20-21, we rephrased from "main driver of anomalously high RIEs are asymmetric ionospheric conditions as only few events" to "main driver of anomalously high RIEs are asymmetric ionospheric conditions and possibly residual error effects from ray tracing through the 3D ionosphere, since only few events"; on p. 7, line 9, we changed from "some perturbations may also come in from" to "some perturbations also come in from"; on p. 15, line 17, we replaced "dominance of asymmetry effects in driving" by "dominance of asymmetry and 3D ray tracing effects in driving the"; and on p. 16, line 6, we updated from "play major roles" to "play important roles".

The authors should consider citing a recent publication that is related to their work: Coleman, C. J., and B. Forte (2017), On the residual ionospheric error in radio occultation measurements, Radio Sci., 47(3), 653–20, doi:10.1002/2016RS006239. Ok, thank you for this suggestion; we have included this citation at two places where we found it fitting, on p.3, line 24, and on p. 5, line 2 (sentence added "Recently also Coleman and Forte (2017) reported RIE investigations for asymmetry conditions, including on the effect of traveling ionospheric disturbances upon the RIE.").

In the spirit of remarks made by co-author Kirchengast at IROWG-6 on referencing unpublished work, the authors might consider referencing the presentation by Mannucci et al., at OPAC-2010, on the role of ionospheric asymmetry in the magnitude of the RIE. If the authors of this paper believe that Mannucci et al.'s presentation is not sufficiently relevant, no reference is needed. Ok, thanks also for this suggestion, we looked into this Mannucci et al. presentation and found it is a good complementary citation. We included it on p. 4, lines 16-19 ("the RIE biases have a clear negative tendency and a magnitude increasing with solar activity as well as are affected by deviations from ionospheric spherical symmetry (Mannucci et al., 2010)"), and on p. 17, lines 5-6 ("strengthening previous results by Mannucci et al. (2010, 2011) we find that asymmetric ionospheric conditions play an important role for anomalously high RIEs").

The authors tend not to emphasize ionospheric asymmetry in their 2015 published study, whereas it factors into this study in a major way. A remark or two on why this study found asymmetry as important given the prior 2015 study is recommended. In our 2015 AMT paper we in fact also briefly discussed the ionospheric asymmetry as a factor of the residual ionospheric errors in bending angles but given that study was focusing on the overall error statistics it was a side issue. We rechecked our description in this study – i.e., how we introduce and discuss how this paper builds on the 2015 paper and why here asymmetry is more in focus etc. (in the introduction, also again in the first paragraphs of summary and conclusions) – and found our relevant statements quite sufficient. We therefore preferred no further changes related to this point.

2 Specific Comments

Detailed Comments:

Page 4, Line 14: The term "quasi-realistic" is somewhat vague. It is understood that simulations are approximations of reality, so what is meant by this phrase? Ok, we decided in this case to just cancel "quasi-realistic" here, since we find the sentence still ok this way.

Page 4, Line 20: I believe unpublished work by Mannucci et al. (OPAC 2010) contained an initial analysis of the role of ionospheric asymmetry. Certainly, the present work is more comprehensive than was presented then. Ok, we agree; see the relevant response above, we included this citation now.

Page 6, Line 16: "clustered" Ok, "cluttered" is not so fitting; we changed to "distributed more diversely" which is better expressing what we intend to express here.

Page 6, Line 17: How is standard deviation defined for the bottom panels of Figure 2? The RIE bias and standard deviation are defined in the same way as their definitions in our 2013 ASR paper, and the Liu et al. ASR 2013 paper has therefore been cited now here ("defined in the same way as by Liu et al. (2013), which are estimated...")

P6, L19: how is spherical symmetry (SS) achieved in the model? Please provide a brief description. In the paragraph just before this one we already have a brief description how this modeling works, and point for further details to "see Liu et al. (2015), Table 2 and Sect. 2.3 therein." Since this cited AMT 2015 paper is open-access/very readily available for all more specifically interested readers we prefer not to have a more detailed description here.

P7, L6: what is meant by "geographical dependencies" as separated from SS conditions or similar? Pure geography (latitude and longitude) would not be a factor, in general. Ok, we have inserted "geographic location dependencies (e.g., via solar or geomagnetic influences), and" so that it is more clear now what is meant.

P11, L19: are out-of-occultation-plane refractive gradients ignored or included? Please clarify. Yes, the gradients are included in 3D, i.e., it is a full-3D ray tracer. We clarified by adding in"3D", i.e., saying now "counts the (numerical 3D ray tracer) points along the ray path..."

P17, L5: to bolster this conclusion and its applicability to real observations, it would be useful to see a metric of ionospheric asymmetry in other regions, to show that in the regions highlighted here, ionospheric asymmetry is largest. If ionospheric asymmetries are large in other regions also, but RIE is not so large, then how robust is this conclusion? See the answer to the major comment 1 above; as we explain there, we have toned down this conclusion so that it now more cautiously reflects the results of this study where we could not (yet) fully separate the iono.asymmetry and technical ray-trace effects.